# Specific and sensitive loop-mediated isothermal amplification (LAMP) method for *Madurella* strains, eumycetoma filamentous fungi causative agent

Isato Yoshioka[1,2], Yugo Mori[1], Ahmed Hassan Fahal[3], Emmanuel Edwar Siddig[3], Satoshi Kaneko[4,5], Takashi Yaguchi [1] *

1 Medical Mycology Research Center, Chiba University, Chiba, Chiba, Japan, 2 Research Institute for Science and Engineering, Waseda University, Tokyo, Japan, 3 Mycetoma Research Centre, University of Khartoum, Khartoum, Sudan, 4 School of Tropical Medicine and Global Health, Nagasaki University, Nagasaki, Japan, 5 Department of Ecoepidemiology, Institute of Tropical Medicine (NEKKEN), Nagasaki University, Nagasaki, Japan

* yaguchi@chiba-u.jp

**Data Availability Statement:** All relevant data are within the paper and/or its Supporting Information files.

## Abstract

### Background

Filamentous fungi of the genus *Madurella* are the primary causative agents of mycetoma, a disease observed in tropical and subtropical regions. Since early diagnostics based on a morphological approach are difficult and have many shortcomings, a molecular diagnostic method suitable for rural settings is required. In this study, we developed the loop-mediated isothermal amplification (LAMP) method to present a foundational technique of the diagnosis of *Madurella* spp. (*M. mycetomatis*, *M. pseudomycetomatis*, *M. tropicana*, and *M. fahalii*), the common causative organisms of eumycetoma.

### Principal findings

We successfully designed a primer pair targeting the rDNAs of three *Madurella* spp. excluding *M. fahalii*, and detected up to 100 fg of genomic DNA extracted from isolates of *M. mycetomatis* and 1 pg of *M. pseudomycetomatis* and *M. tropicana*, within one hour. Second, a primer pair specific to *M. mycetomatis*, the most common causative species, or *M. fahalii*, a drug-resistant species, was constructed, and the detection limit of both primer pairs was 1 pg. The designed primers accurately distinguished 16 strains of the genus *Madurella* from various fungal species known to cause mycetomas.

### Conclusion

In summary, we established the first model of a LAMP detection method that rapidly and sensitively detects and identifies *Madurella* isolates for clinical diagnostics. Moreover, the combined designed primer sets could identify mycetoma-causing strains simultaneously.

**Funding:** Japan Agency for Medical Research and Development (AMED) under Grant Number JP21jm0510005, SK. The funders had no role in study design, data collection and analysis, decision to publish, or preparation of the manuscript.

**Competing interests:** The authors have declared that no competing interests exist.

## Author summary

Mycetoma is known as a neglected tropical disease causing chronic granulomatous inflammation of the subcutaneous tissue due to infection by microorganisms. In particular, a simple molecular diagnostic method for fungal mycetoma (eumycetoma) is required for a community-based medical treatment, because of the difficulty in isolating and identifying the causative fungus. In this study, we established the loop-mediated isothermal amplification (LAMP) method to identify *Madurella* strains which are common causative agents of eumycetoma. The results suggest that diagnosis using the LAMP reaction enables specific and sensitive detection of *Madurella* by incubating the sample at a constant temperature and visually confirming DNA amplification. Based on these results, we expect that the LAMP-based diagnosis will contribute to the treatment of eumycetoma in rural settings

## Introduction

Mycetoma is a chronic granulomatous inflammatory disease of the subcutaneous tissue classified as bacterial (actinomycetoma) or fungal (eumycetoma) according to the causative agents. It is endemic to tropical and subtropical areas (the mycetoma belt) [1,2]. Especially, eumycetoma has been frequently reported in African countries [2,3]. Since mycetoma in the late stage leads to massive deformities and disabilities and sometimes becomes fatal, a rapid and proper diagnosis is essential [4,5]. *Madurella mycetomatis* and related species are common causative agents of eumycetoma; thus, they are important targets of diagnosis [2,3,6,7]. Morphological identification of *Madurella* spp. is difficult because culture media lacks sporulation and has a slow growth rate [8]. These circumstances have become a severe diagnostic problem [6,9].

Molecular diagnostic methods have been developed for the rapid and accurate detection of *Madurella* spp. in mycetoma specimens [9]. In 1999, Ahmed et al. designed PCR primer pairs that specifically target the rDNA-ITS region of *Madurella* spp. [10,11]. Recently, it was confirmed that not only *M. mycetomatis* but also other *Madurella* species, including *M. fahalii*, an itraconazole-resistant agent [7,12,13], can cause eumycetoma [13,14]. Therefore, species-specific detection of causative *Madurella* spp. is essential. Species-specific PCR methods have improved the detection of *M. mycetomatis* [11] and *M. pseudomycetomatis* [15]. Furthermore, a real-time PCR method, which could distinguish *M. mycetomatis* and *M. fahalii* from other strains, such as *M. pseudomycetomatis* and *M. tropicana*, was developed to exclude false-positive amplification based on the melting points of products and Ct values [16].

A community-based diagnostic test is required for the early detection and management of mycetoma in rural settings and its epidemiologic study. [4,5,17]. From this viewpoint, the isothermal DNA amplification technique is preferred because it does not require expensive instruments or skilled laboratory personnel compared with the PCR method [18,19]. Rolling circle amplification (RCA), recombinase polymerase amplification (RPA), and loop-mediated isothermal amplification (LAMP) have been developed to detect *M. mycetomatis* [18,20]. LAMP is the best choice among the isothermal amplification techniques due to its sensitivity and specificity [21,22]. Moreover, this is a cost-effective, well-developed, and versatile detection method [18,19].

In this study, we developed a LAMP method that targets four known *Madurella* spp. (*M. mycetomatis*, *M. pseudomycetomatis*, *M. tropicana*, and *M. fahalii*), which are the most frequent causative organisms for rapid, accurate, and versatile diagnostic methods for mycetoma.

## Materials and methods

### Fungal and bacterial strains

A total of 34 fungal- and 5 bacterial strains were used in this study (Table 1). The details of fungal strains are as follows: 16 belong to *Madurella*, three to other members of *Chaetomiaceae*, four to *Aspergillus*, and 11 to other taxa recognized as causative agents of mycetoma [3,11]. The species of five bacterial strains used in this study are also known to cause actinomycetoma [2]. Briefly, 14 clinical strains were preserved at Medical Mycology Research Center (MMRC; Chiba University, Japan) as IFM collection through the National Bio-Resource Project, Japan, and 13 mycetoma strains preserved at the Mycetoma Research Center (MRC) Khartoum University, Khartoum, Sudan. In addition, 12 strains were obtained from the culture collections of the Westerdijk Fungal Biodiversity Institute (CBS; Netherlands), NITE Biological Resource Center (NBRC; Japan), American Type Culture Collection (ATCC; United States) and Japan Collection of Microorganisms (JCM; Japan). The MRC strains were brought to Chiba University as slant-cultured samples, and their genomic DNAs were extracted after subculturing, as described below. Several DNA samples were purchased from CBS (two samples) and brought from MRC (five samples), and tested in Japan (Table 1).

### Preparation of genomic DNA

All strains were grown on a PDA medium at 25°C for 1–3 weeks. Bacterial colonies or mycelia were picked by inoculating the needle and suspended in 500 μL of 100 mM Tris-HCl buffer (pH 8.5) containing 40 mM EDTA. The bacterial cell or fungus body in the suspension was disrupted using MagNA Lyser and MagNA Lyser Green Beads (Roche Diagnostics, Indianapolis, IN) at 5000 rpm for 20 s. Genomic DNAs were extracted using the benzyl chloride method [23] with some modifications. Finally, the concentration of DNA solution was determined by absorbance at 260 nm using microvolume spectrophotometer DS-11 (Denovix, Wilmington, DE) and the solution was stored at 4°C in Tris-EDTA buffer.

### Design of LAMP primers

Three primer pairs were designed: one targeting three *Madurella* strains (*M. mycetomatis*, *M. pseudomycetomatis* and *M. tropicana*), another specifically for *M. mycetomatis and* the third for *M. fahalii*. Candidate primer pairs, including loop primers [22], were obtained from the rDNA sequences of *M. mycetomatis* CBS 109801[T] (DDBJ/EMBL-Bank/GenBank accession no. DQ836767.1), *M. pseudomycetomatis* CBS 129177[T] (accession no. EU815933.1), *M. tropicana* CBS 201.38[T] (accession no. JX280869.1), and *M. fahalii* CBS 129176[T] (accession no. JN573178.1) using Primer Explorer V5 (http://primerexplorer.jp/lampv5e/index.html; Eiken Chemical Co. Ltd., Tokyo, Japan). For primers toward three *Madurella* strains, the nucleotide mismatches among them were allowed, and the sequence of *M. mycetomatis* was preferentially used. Similarly, primer pairs specific for *M. mycetomatis* and *M. fahalii* were generated. In this study, the primer sets targeting (1) three *Madurella* spp., (2) only *M. mycetomatis* and (3) only *M. fahalii* were termed as (1) PV, (2) PM and (3) PF, respectively. The sequences of the primer pairs targeting the three *Madurella* strains, *M. mycetomatis*, and *M. fahalii*, are listed in Table 2.

### LAMP amplification of fungal DNA

The LAMP reaction was performed in a 25 μL reaction volume by dissolving the dried LAMP reagent in the primer solution and DNA solution and diluting it with 10 mM Tris-HCl buffer (pH 8.0). An aliquot (15 μL) of the primer solution containing 40 pmol FIP and BIP, 20 pmol

**Table 1. Fungal and bacterial strains used in this study.**

| Species | Strain No.[1] | Note |
|---|---|---|
| **Fungi** | | |
| *Madurella mycetomatis* | IFM 46458 | |
| | IFM 68169<br>(= MRC No. 6) | |
| | IFM 68173<br>(= MRC No. 16) | |
| | IFM 68177<br>(= MRC No. 31) | |
| | IFM 68178<br>(= MRC No. 33) | |
| | MRC 59v7 | The extracted DNA was brought to Japan from Sudan |
| | MRC 68v7 | The extracted DNA was brought to Japan from Sudan |
| | MRC 73v7 | The extracted DNA was brought to Japan from Sudan |
| | MRC 88v7 | The extracted DNA was brought to Japan from Sudan |
| | MRC 89v7 | The extracted DNA was brought to Japan from Sudan |
| *Madurella pseudomycetomatis* | IFM 46460 | |
| *Madurella tropicana* | CBS 201.38$^T$ | The extracted DNA was purchased from CBS |
| *Madurella fahalii* | CBS 129176$^T$ | The extracted DNA was purchased from CBS |
| | IFM 68170<br>(= MRC No. 9) | |
| | IFM 68171<br>(= MRC No. 13) | |
| | IFM 68242<br>(= MRC No. 25) | |
| *Aspergillus fumigatus* | IFM 67766 | |
| *Aspergillus niger* | IFM 68248 | |
| *Aspergillus nidulans* | IFM 68245<br>(= MRC No. 4) | |
| *Aspergillus terreus* | IFM 67509 | |
| *Chaetomium globosum* | CBS 148.51$^T$ | |
| *Chaetomium rectangulare* | CBS 126778$^T$ | |
| *Curvularia lunata* | IFM 64654 | |
| *Exophiala jeanselmei* | IFM 67393 | |
| *Falciformispora senegalensis* | CBS 196.79$^T$ | |
| *Falciformispora tompkinsii* | CBS 200.79 | |
| *Fusarium oxysporum* | IFM 61457 | |
| *Fusarium solani* | IFM 67677 | |
| *Neocosmospora falciformis*<br>(≡ *Acremonium falciforme*) | CBS 475.67 | |
| *Neotestudina rosatii*<br>(≡ *Zopfia rosatii*) | CBS331.78 | |
| *Sarocladium kiliense*<br>(≡ *Acremonium kiliense*) | CBS 158.61 | |
| *Scedosporium apiospermum* | IFM 66388 | |
| *Thermothielavioides terrestris*<br>(≡ *Thielavia terreistris*) | CBS 355.66<br>(= NBRC 9121) | |
| *Trichophyton rubrum* | IFM 66913 | |
| **Bacteria** | | |
| *Actinomadura madurae* | IFM 0585$^T$<br>(= JCM 7436$^T$) | |

(*Continued*)

**Table 1.** (Continued)

| Species | Strain No.[1] | Note |
|---|---|---|
| *Actinomadura pelletieri* | IFM 0590[T]<br>(= JCM 3388[T]) | |
| *Nocardia asteroides* | IFM 10791 | |
| *Nocardia brasiliensis* | IFM 12361 | |
| *Streptomyces somaliensis* | IFM 11185 | |

LF and LB, and 5 pmol F3 and B3 was added to a reaction tube containing Loopamp DNA Amplification Reagent D (Eiken Chemical Co., Ltd.) and mixed with 10 μL template DNA solution using a centrifuge machine equipped with a shaking function (MS-16; Eiken Chemical Co., Ltd.) to dissolve the dried LAMP reagent in the tube. The resultant mixture was immediately transferred to a LAMP instrument (LoopampEXIA (Eiken Chemical)), and its turbidity, reflecting DNA amplification, was measured in real-time for 60 min at a constant temperature. Unless otherwise stated, the reaction was performed at 63˚C. To verify the specificity and versatile reactivity to *Madurella* strains, 10 μL of 1 ng/μL genomic DNA solution (10 ng of genomic DNA) was used per reaction tube. In contrast, to determine the detection limit, 10 μL of 1 ng/L–1 fg/L (10 ng–10 fg DNA) solution was tested.

**Table 2. Primers used in this study.**

| Primer set | Region | Sequence (5′ to 3′) |
|---|---|---|
| PV | FIP (F1c/F2) | ACCAAGAGATCCGTTGTTGAAAGT–CCGGAGGATTATACAACACCC |
| | BIP (B1c/B2) | CTGGCATCGATGAAGAACGCAG–GGCGCAATGTGCGTTCAAAG |
| | F3 | CTCCCCAAACCATTGTGAACATAC |
| | B3 | CATGCCCGCCAGAATACTGG |
| | LF | ACTCAGAGAGGCCGTACAGAGC |
| | LB | TGCAGAATTCAGTGAATCATCG |
| PV2 | FIP (F1c/F2) | CGGACAGCCGCAGGTCCC–GGCATGCCTGTTCGAGC |
| | BIP (B1c/B2) | CCAGTGGCGGGCTCGCTG–ACGCCCTGGGCGAGTTG |
| | F3 | CGAATCTTTGAACGCACATTG |
| | B3 | GATCCGAGGTCAACCTTGG |
| | LF | GGGGCTTGATGGTTGAAATGAC |
| | LB | GTCACCCCGAGCGTAGTAGT |
| PM | FIP (F1c/F2) | GGACACTACACTACCGGGAGGC–ATACCCCCAAAACCGTTGC |
| | BIP (B1c/B2) | CACCCTATTTGCTCTGTACGGCC–CCAAGAGATCCGTTGTTGAAAG |
| | F3 | ACTCCCCAAACCATTGTGAAC |
| | B3 | ATTACTTATCGCATTTCGCTGC |
| | LF | GTGCCGCCCGCCGAA |
| | LB | CTCTGAGTCTTCTGTACTGAATAAGTC |
| PF | FIP (F1c/F2) | CAACCCGAGAGACCCGAGAG–AACCATTGTGAACCTACCCA |
| | BIP (B1c/B2) | ACCCTGTACTTTGTATGGCCTCT–ACCAAGAGATCCGTTGTTGA |
| | F3 | ACAGAGTTGCAAAACTCCCA |
| | B3 | CGCTGCGTTCTTCATCGA |
| | LF | CGCCGAAGCAACGGTT |
| | LB | CTGAGTCTCTGTACTGAATAAGTCA |

## Results

### Design of the primer pairs detecting *Madurella* spp

We constructed a primer set targeting the four strains of *Madurella* based on their rDNA sequences. A negative control strain was searched using NCBI BLAST with the rDNA sequence of *M. mycetomatis* as the query, and *Chaetomium rectangulare* (accession no. NR_144817) showed the highest similarity among the strains preserved in the MRC. Therefore, *C. rectangulare* CBS 126778[T] was selected as the negative control to validate specificity, though there is no report that it causes mycetoma. Nucleotide alignment of partial rDNA sequences derived from the four *Madurella* species and *C. rectangulare* was conducted to design primers, as shown in Fig 1. However, the target region is conserved among these species

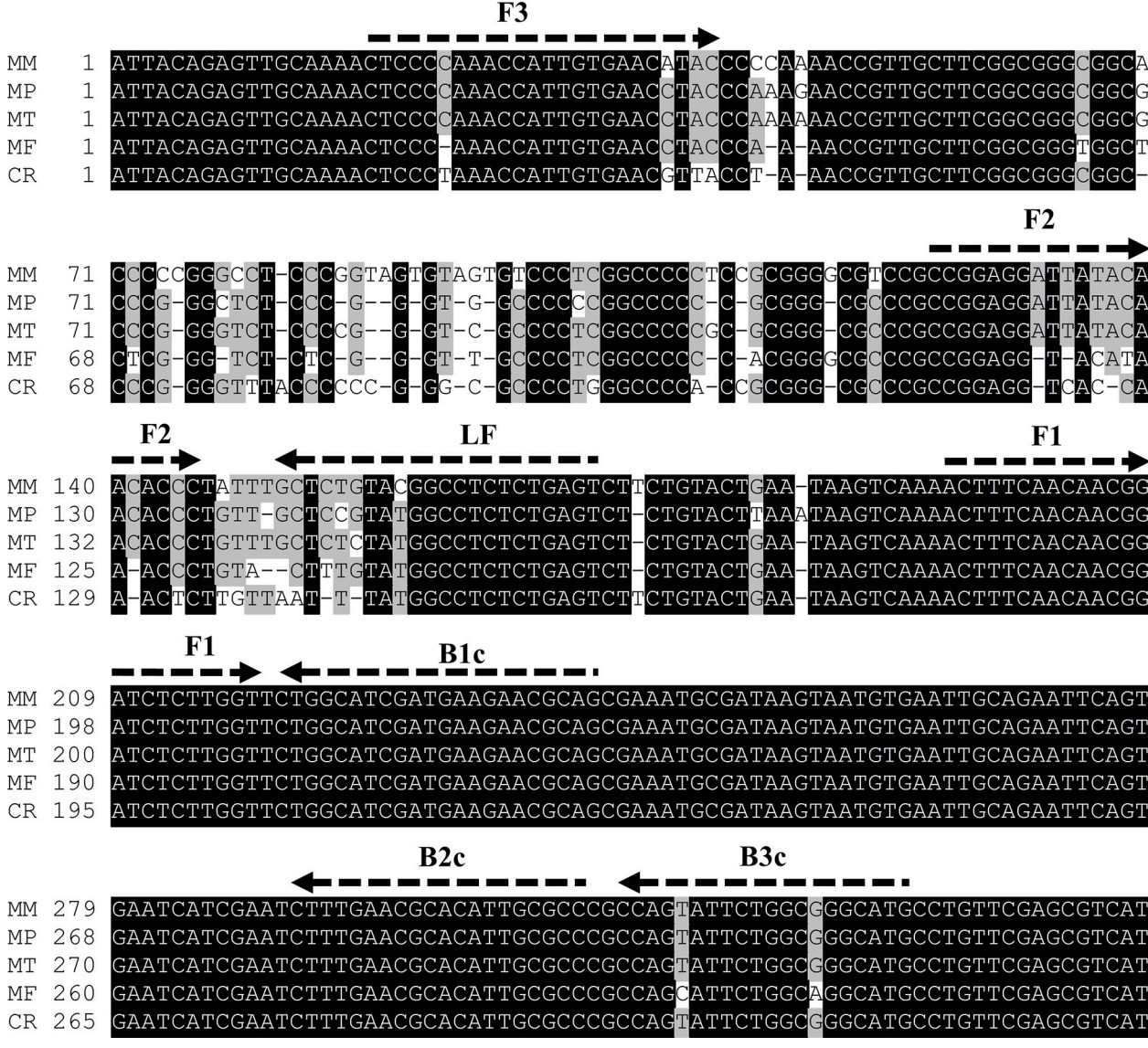

**Fig 1. Design of primer targeting three *Madurella* spp. (primer PV).** The alignment of rDNA sequences among *Madurella* strains and *Chaeromium rectangluare* (negative control) is shown. Arrows illustrate the primers' positions. The species name and the number of the first nucleotide are represented at each line using the abbreviations as followings: MM, *M. mycetomatis* (accession no. DQ836767.1); MP, *M. pseudomycetomatis* (accession no. EU815933.1); MT, *M. tropicana* (accession no. MK926825.1); MF, *M. fahalii* (accession no. JN573178.1); CR, *Chaetomium rectangluare* (NR_144817). The nucleotide alignment and illustration are performed using Genetyx Ver. 14 (Genetyx Co., Tokyo, Japan) with its default parameters.

but is different from those of other fungi. Thus, we designed a primer pair targeting the conserved regions in three species (*M. mycetomatis*, *M. pseudomycetomatis*, and *M. tropicana*) harboring large mismatches in the F2 and F3 regions of *M. fahalii* and *C. rectangulare* (primer set PV). To detect all known *Madurella* spp., the primers targeting *M. fahalii* have also been developed. In addition, we established the specific amplification of the major causative agent, *M. mycetomatis*. Based on the nucleotide alignment with the rDNA sequence derived from *M. pseudomycetomatis*, we developed species-specific primers for *M. mycetomatis* (primer set PM) and *M. fahalii* (primer set PF) as shown in Fig 2A and 2B, respectively.

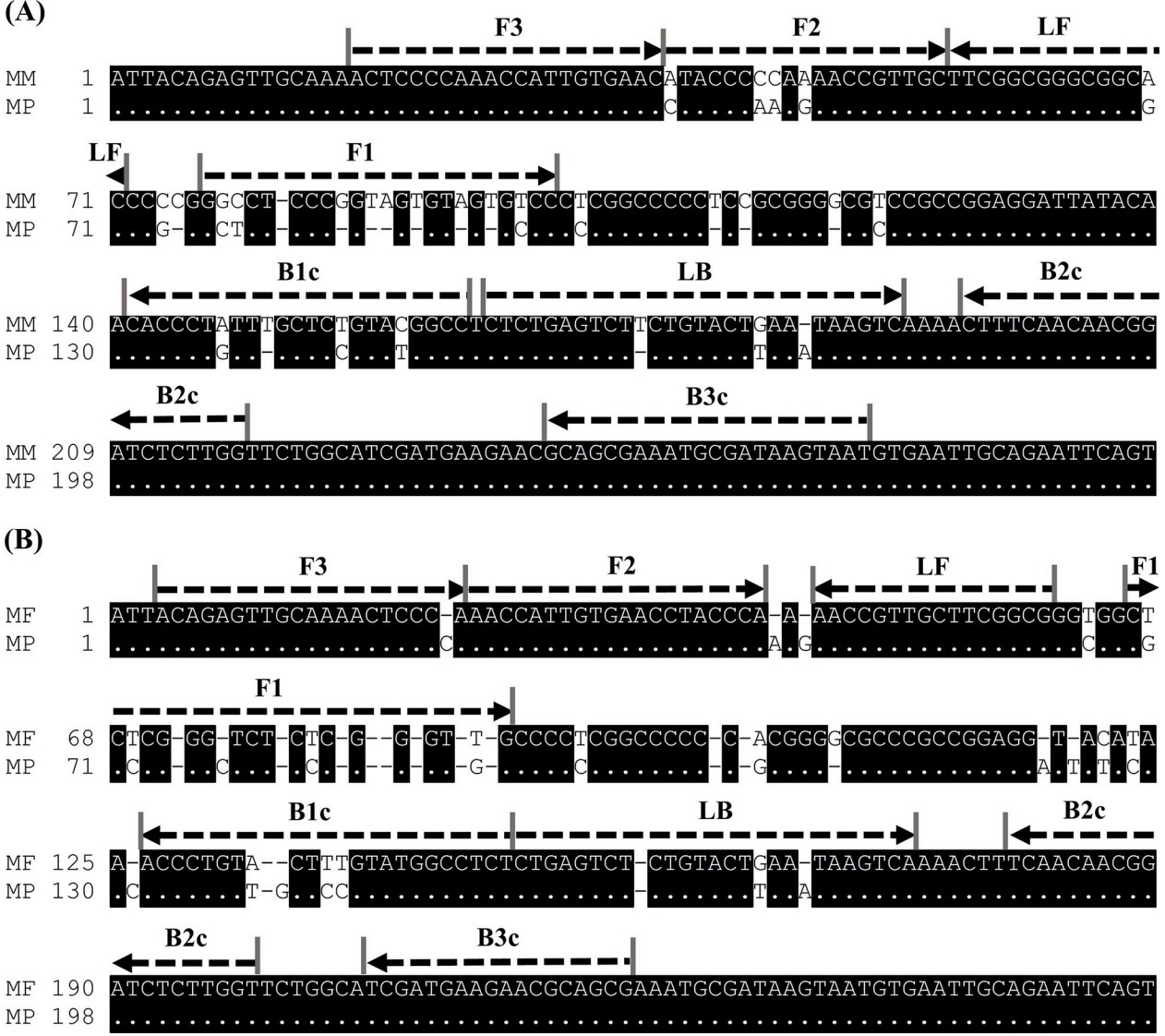

**Fig 2. The primers' design specific to *M. mycetomatis* (primer PM) or *M. fahalii* (primer PF).** The rDNA sequence of *M. mycetomatis* (A) or *M. fahalii* (B) was aligned with that of *M. pseudomycetomatis* (accession no. EU815933.1), and arrows represented the primer positions. The matched nucleotide position was represented using dot signs, while the gap and mismatch positions using corresponding substitution ones. The species name and the number of first nucleotide were represented at each line using the following abbreviations: MM, *M. mycetomatis*; MP, *M. pseudomycetomatis*; MF, *M. fahalii*. The nucleotide alignment and illustration were performed using Genetyx Ver. 14 (Genetyx Co., Tokyo, Japan) with its default parameters.

### Validation of LAMP primers

The optimal reaction temperature for each primer was determined. For PV and PM, 100 pg of genomic DNA derived from *M. mycetomatis* IFM 46458 was used. In contrast, for PF, 100 pg of genomic DNA of *M. fahalii* CBS 129176$^{\text{T}}$ was used. As shown in S1A Fig, PV showed the comparable amplification rates at the reaction temperature between 62˚C and 65˚C, with decreased efficiency at 61˚C or 66˚C. Similarly, the amplification rates using PM and PF reached a maximum at 63˚C, as shown in S2 and S3 Figs. Therefore, the optimal reaction temperature for all three primer sets was determined as 63˚C.

To assess the specificity of these primers for *Madurella* spp., we performed a LAMP reaction at 63˚C using a sufficient amount (10 ng) of the genomic DNAs derived from *M. mycetomatis* IFM 46458, *M. pseudomycetomatis* IFM 46460, *M. tropicana* CBS 201.38$^{\text{T}}$, *M. fahalii* CBS 129176$^{\text{T}}$, and *C. rectangulare* CBS 126778$^{\text{T}}$ as templates. As expected, PV detected *M. mycetomatis*, *M. pseudomycetomatis*, and *M. tropicana* and showed no DNA amplification using the genomes of *M. fahalii* and *Chaetomium rectangulare*, as shown in Fig 3A. Additionally, PM and PF specifically amplified the genomic DNA of *M. mycetomatis* and *M. fahalii*, as

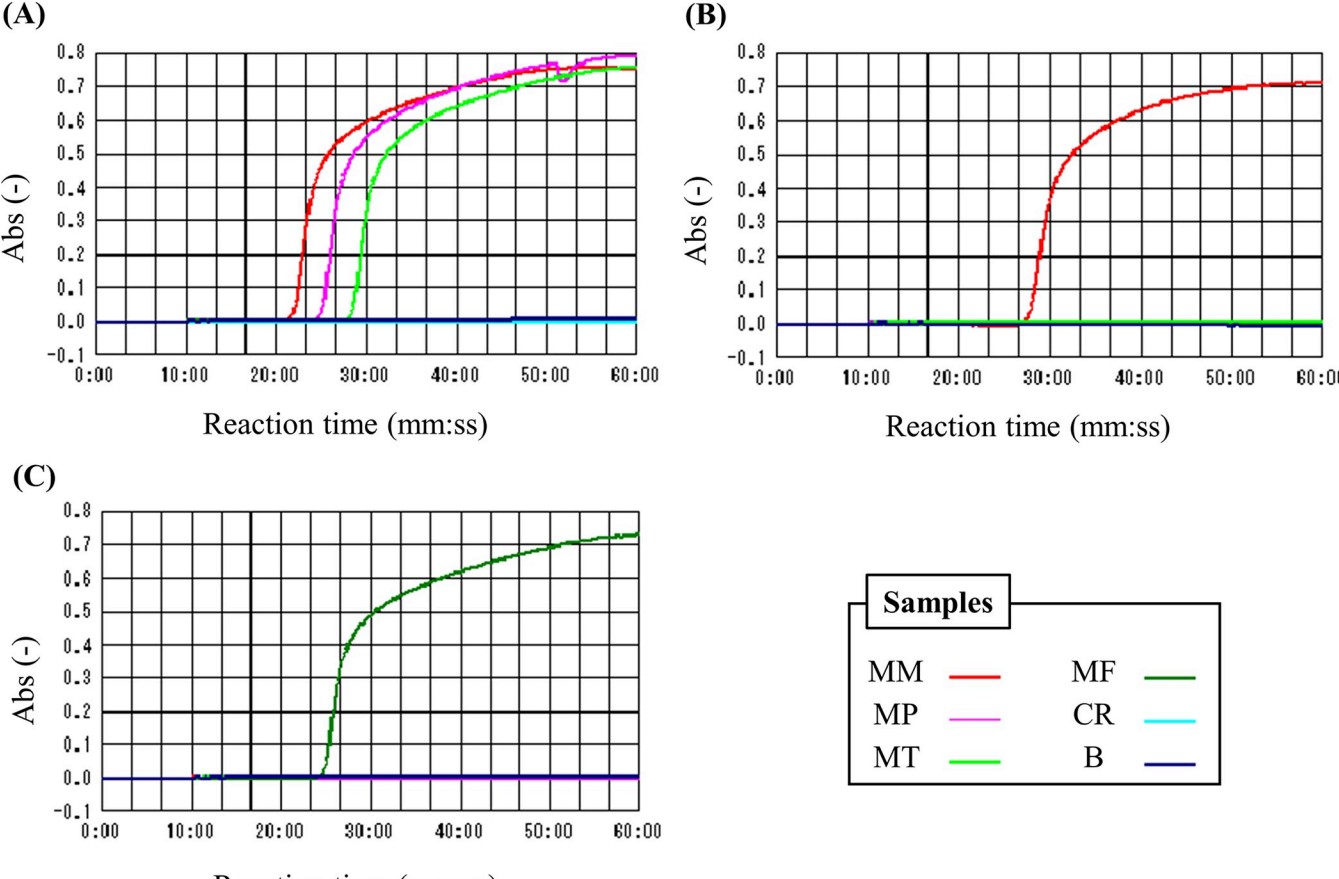

**Fig 3.** The specificity of designed LAMP primers, (A) PV, (B) PM and (C) PF. 10 ng of the extracted genomic DNAs derived from *Madurella* spp. were used as templates. LAMP reaction was performed at 63˚C for 60 min, and the turbidity of the reaction solution was monitored as an indicator of amplification. As for primer PV, the genomic DNA of *Chaetomium rectangulare* was also used. The sample legends of the plots were represented on the bottom right panel. Abbreviations: MM, *M. mycetomatis* IFM 46458; MP, *M. pseudomycetomatis* IFM 46460; MT, *M. tropicana* CBS201.38$^{\text{T}}$; MF, *M. fahalii* CBS129176$^{\text{T}}$; CR, *Chaetomium rectangluare* CBS 126778$^{\text{T}}$; B, Blank sample with a buffer used for DNA dilution.

shown in Fig 3B and 3C, respectively. On the other hands, we also designed the primer pair (primer set PV2) based on the ITS-2 region, which has potential to amplify all four *Madurella* spp. due to its highly conserved sequence, as shown in S4 Fig. As expected, primer PV2 could amplify the genomic DNAs of four *Madurella* spp., but it also showed the reactivity to *C. rectangulare*, at an optimal reaction temperature (65˚C). Thus, we performed further experiments using the primers PV, PM and PF as basal primer sets.

The sensitivity of the primer sets was determined by amplifying the genome in different amounts, ranging from 10 fg to 10 ng. PV could detect up to 100 fg genomic DNA of *M. mycetomatis* and *M. pseudomycetomatis* but failed to do for *M. tropicana*, as shown in Fig 4. However, the 1 pg genome of these three species was successfully amplified, as shown in S4B Fig. In addition, PM and PF recognized up to 1 pg of genomic DNAs derived from *M. mycetomatis* and *M. fahalii*, respectively, as shown in Fig 4B and 4C. Finally, the specificity of the LAMP reaction was verified using agarose gel electrophoresis. As shown in S5 Fig, the resultant solution in the positive samples showed the typical ladder bands while none of amplicons were detected in the reaction using non-target sample. These results suggest that the expected amplification occurred.

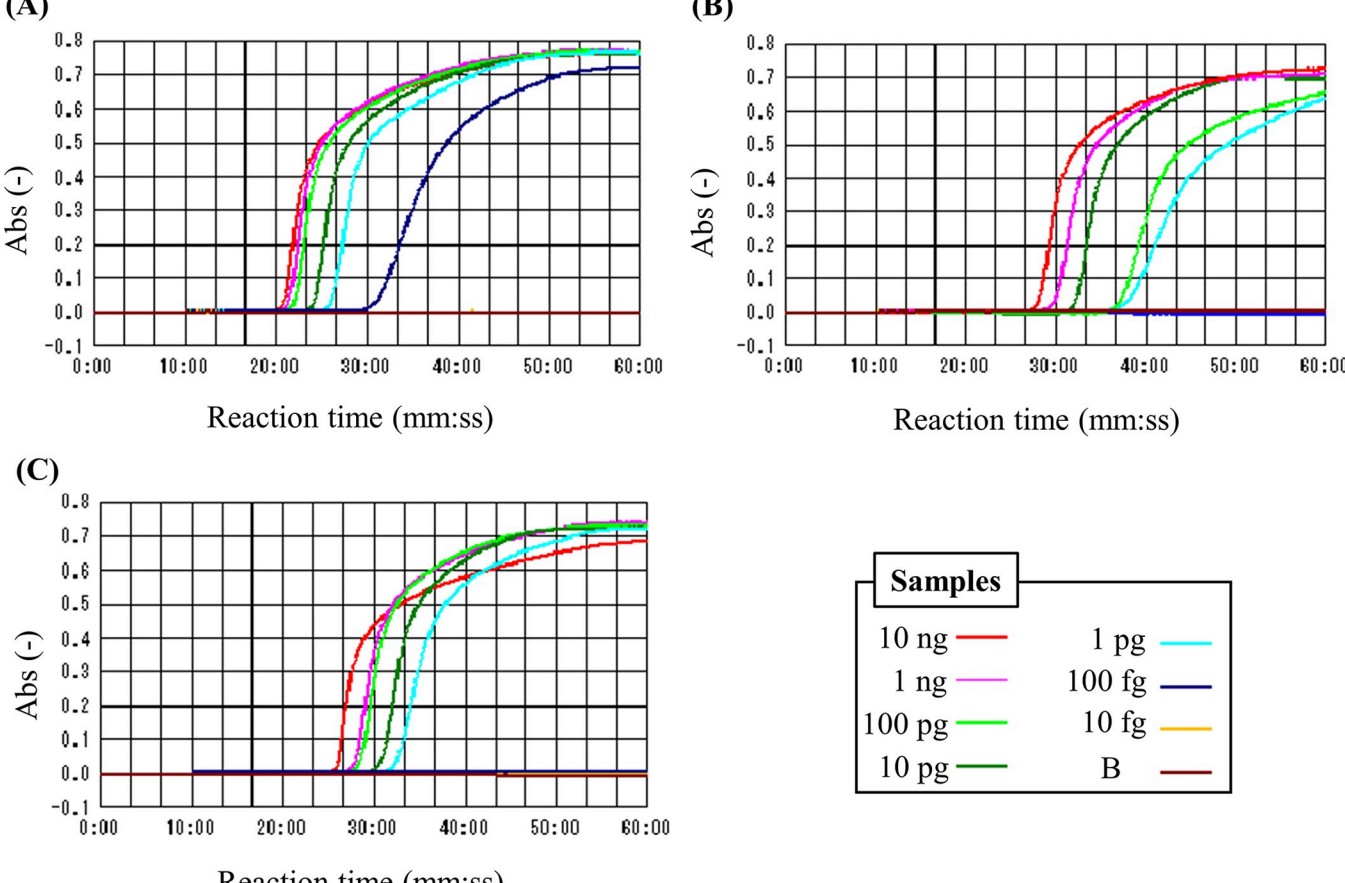

**Fig 4.** The detection limits of designed LAMP primers, (A) PV, (B) PM and (C) PFThe genomic DNAs of *M. mycetomatis* IFM 46458 (for PV and PM) and *M. fahalii* CBS 126778[T] (for PF) were serially diluted by 10-fold, and 10 ng—10 fg of DNAs were used as a template. LAMP reaction was performed at 63˚C for 60 min, and the turbidity of the reaction solution was monitored as an indicator of amplification. The sample legends of the plots were represented on the bottom right panel, and a plot labelled with B corresponds to the data of a blank sample with a buffer used for DNA dilution.

**Table 3. LAMP amplification for clinical isolates and pathogenic fungi and bacteria, which are known as causative agents of mycetoma.**

| Species[1] | Primer set (positive samples / tested samples)[2,3] | | |
|---|---|---|---|
| | PV | PM | PF |
| *Madurella mycetomatis* | + (10/10) | + (10/10) | - (0/10) |
| *Madurella pseudomycetomatis* | + (1/1) | - (0/1) | - (0/1) |
| *Madurella tropicana* | + (1/1) | - (0/1) | - (0/1) |
| *Madurella fahalii* | - (0/4) | - (0/4) | + (4/4) |
| *Aspergillus fumigatus* | - (0/1) | - (0/1) | - (0/1) |
| *Aspergillus niger* | - (0/1) | - (0/1) | - (0/1) |
| *Aspergillus nidulans* | - (0/1) | - (0/1) | - (0/1) |
| *Aspergillus terreus* | - (0/1) | - (0/1) | - (0/1) |
| *Chaetomium globosum* | - (0/1) | - (0/1) | - (0/1) |
| *Curvularia lunata* | - (0/1) | - (0/1) | - (0/1) |
| *Exophiala jeanselmei* | - (0/1) | - (0/1) | - (0/1) |
| *Falciformispora senegalensis* | - (0/1) | - (0/1) | - (0/1) |
| *Falciformispora tompkinsii* | - (0/1) | - (0/1) | - (0/1) |
| *Fusarium oxysporum* | - (0/1) | - (0/1) | - (0/1) |
| *Fusarium solani* | - (0/1) | - (0/1) | - (0/1) |
| *Neocosmospora falciformis* (≡ *Acremonium falciforme*) | - (0/1) | - (0/1) | - (0/1) |
| *Neotestudina rosatii* (≡ *Zopfia rosatii*) | - (0/1) | - (0/1) | - (0/1) |
| *Sarocladium kiliense* (≡ *Acremonium kiliense*) | - (0/1) | - (0/1) | - (0/1) |
| *Scedosporium apiospermum* | - (0/1) | - (0/1) | - (0/1) |
| *Thermothielavioides terrestris* (≡ *Thielavia terreistris*) | - (0/1) | - (0/1) | - (0/1) |
| *Trichophyton rubrum* | - (0/1) | - (0/1) | - (0/1) |
| *Actinomadura madurae* | - (0/1) | - (0/1) | - (0/1) |
| *Actinomadura pelletieri* | - (0/1) | - (0/1) | - (0/1) |
| *Nocardia asteroides* | - (0/1) | - (0/1) | - (0/1) |
| *Nocardia brasiliensis* | - (0/1) | - (0/1) | - (0/1) |
| *Streptomyces somaliensis* | - (0/1) | - (0/1) | - (0/1) |

## LAMP amplification using known causative species

To verify the availability of the designed primers, we performed a LAMP reaction using 10 ng DNA from mycetoma clinical isolates and species known as the mycetoma causative agents including bacteria [3,11,16]. The PV amplification test detected only ten strains of *M. mycetomatis*, *M. pseudomycetomatis*, and *M. tropicana* (Table 3). Furthermore, PM and PF completely distinguished their targets from those of the other *Madurella* species.

## Discussion

Mycetoma is a severe deep mycosis and is widely recognized as one of the most neglected tropical disease [1]. Globally, *Madurella* spp. are the most common causative agents of eumycetomas. Due to the challenges in isolating and morphologically identifying *Madurella* in culture media [4,9], many studies have reported using molecular techniques for diagnosing mycetoma [11,16,18,24]. Since isothermal DNA amplification techniques are regarded as a promising tool in community-based medical treatment, this study aimed to develop a LAMP method for *Madurella* spp.. Three primer sets were designed to target three *Madurella* spp.: one for each

*M. mycetomatis* and *M. fahalii*. Finally, these primers enabled a LAMP method which specifically amplified a small quantity of genomic DNAs derived from *Madurella* spp. using a constant temperature.

Combining the designed primers simultaneously and specifically identified four common eumycetoma-causing organisms; *M. mycetomatis*, *M. fahalii*, *M. pseudomycetomatis*, and *M. tropicana* at the same reaction temperature, i.e. in one batch reaction, as shown in Fig 3 and Table 3. Additionally, a double check based on the specific detection of *M. mycetomatis* is also important for rapid downstream treatment since *M. mycetomatis* is the primary fungus causing eumycetoma [2,6], and isolates with reduced susceptibility to azoles have emerged [25]. Similarly, the double check for *M. fahalii* will become significant because it is known to exhibit azole-resistance. To overcome this problem, the primer set targeting all *Madurella* spp. is needed. Unfortunately, such primer sets were not obtained as far as the rDNA ITS regions were used as target, as shown in Figs 4A and S4. On the other hand, *M. pseudomycetomatis* and *M. tropicana* have only been tested in single strains each. Thus, a larger number of strains belonging to these species should be tested in the future to verify the specificity of these primers.

This study reported that LAMP system could detect up to 1 pg genomic DNA of *Madurella* spp. within 60 min, as shown in Fig 4. Based on the estimated molecular number of genomic DNA of *M. mycetomatis*, which is calculated to be 26.6 copies/pg, assuming a genome size of 36.7 Mbp [26]. The system detection limits are comparable not only to those targeting other fungi, such as *Aspergillus* [27] and *Fusarium* [28] but also to those of real-time PCR methods for *Madurella* (up to 3 pg) [16]. Furthermore, the sensitivity of our LAMP system is approximately 200 times higher than that reported previously [18]. This may be because the loop primer drastically enhances reaction efficiency [22]. We improved the *M. mycetomatis*-specific detection method in this study and extended the target species described above.

Recently, real-time PCR has been reported to be a reliable diagnostic tool for mycetoma because the products' Ct value (cycle to amplify) and melting point show clear false positives and species identification criteria by monitoring a fluorescence thermal cycler [16]. On the other hand, the data obtained from this study suggest that the LAMP method can also identify *Madurella* spp. using small amount of DNA (>1 pg) at the endpoint in one batch. Thus, the LAMP method might be an alternative to PCR as a diagnostic method for mycetoma in community-based medicine.

It has been demonstrated that the LAMP reaction can be used with the crude fungal DNA extracted from clinical specimens [29–31], infected plants, or a few conidia with a simple treatment [28,32] due to its outstanding sensitivity and specificity. In addition, the preparation of crude DNA derived from mycetoma specimens for PCR diagnosis has been reported [13,16]. These findings support the idea that our LAMP system will be applicable to clinical samples. However, in this study, specimens of mycetoma were not used and the LAMP reaction was performed only in the laboratory. Thus, the further study in clinical sites is required for practical use of this technique.

In conclusion, the present study showed that the isothermal DNA amplification by LAMP method could identify *Maderalla* spp. as the major causative agent of eumycetoma worldwide. This method is simple, quick, reliable, affordable, cost-effective, and field-friendly. It can replace PCR-based diagnostic tests in low-resource communities and community-based mycetoma epidemiological surveys.

## Supporting information

**S1 Fig. Optimization of LAMP reaction temperature using primer PV.** LAMP reaction was performed using genomic DNA of *Madurella myceromatis* IFM 46458 at the reaction

temperature between 61˚C and 66˚C. The solid line and dotted line represent the data of the genomic DNA sample and blank using a buffer used for DNA dilution, respectively.
(TIF)

**S2 Fig. Optimization of LAMP reaction temperature using primer PM.** LAMP reaction was performed using genomic DNA of *Madurella myceromatis* IFM 46458 at the reaction temperature between 62˚C and 65˚C. The solid line and dotted line represent the data of the genomic DNA sample and blank using a buffer used for DNA dilution, respectively.
(TIF)

**S3 Fig. Optimization of LAMP reaction temperature using primer PF.** LAMP reaction was performed using genomic DNA of *Madurella fahalii* CBS 129176 at the reaction temperature between 62˚C and 65˚C. The solid line and dotted line represent the data of the genomic DNA sample and blank using a buffer used for DNA dilution, respectively.
(TIF)

**S4 Fig. The primers' design targeting all four Madurella spp. (primer PV2).** (A) The alignment of rDNA sequences among *Madurella* strains and *Chaeromium rectangluare* (negative control) is shown. Arrows illustrate the primers' positions. (B) The specificity of primer PV2. LAMP reaction was performed using genomic DNAs derived from *Madurella* spp. and *C. rectangulare* at 65˚C. Abbreviations: MM, *M. mycetomatis*; MP, *M. pseudomycetomatis*; MT, *M. tropicana*; MF, *M. fahalii*; CR, *Chaetomium rectangluare*; B, Blank sample with a buffer used for DNA dilution.
(TIF)

**S5 Fig. Verification of LAMP amplicons using primer sets PM, PM and PF by agarose electrophoresis.** (A) PM (Lanes 1–9) and (B) PM (Lanes 10–13) and PF (Lanes 14–17) by agarose electrophoresis. Lanes: M, EZ load 100 bp Molecular Ruler (Bio-rad, Hercules, CA); 1, *Madurella mycetomatis* IFM 46458 (10 ng); 2, *M. pseudomycetomatis* IFM 46460 (10 ng); 3, *M. tropicana* CBS 201.38 (10 ng); 4, *M. fahali*i CBS 129176 (10 ng); 5, *Chaetomium rectangulare* CBS 126778 (10 ng); 6, *M. mycetomatis* IFM 46458 (1 pg); 7, *M. pseudomycetomatis* IFM 46460 (1 pg); 8, *M. tropicana* CBS 201.38 (1 pg); 9, no template; 10, *M. mycetomatis* IFM 46458 (10 ng); 11, *M. mycetomatis* IFM 46458 (1 pg); 12, *M. fahalii* CBS 129176 (10 ng); 13, no template; 14, *M. fahalii* CBS 129176 (10 ng); 15, *M. fahalii* CBS 129176 (1 pg); 16, *M. mycetomatis* IFM 46458 (10 ng); 17, no template.
(TIF)

## Author Contributions

**Conceptualization:** Ahmed Hassan Fahal, Satoshi Kaneko.

**Data curation:** Isato Yoshioka, Yugo Mori, Takashi Yaguchi.

**Formal analysis:** Yugo Mori.

**Funding acquisition:** Satoshi Kaneko.

**Investigation:** Isato Yoshioka, Yugo Mori, Emmanuel Edwar Siddig, Takashi Yaguchi.

**Methodology:** Isato Yoshioka, Takashi Yaguchi.

**Project administration:** Ahmed Hassan Fahal, Satoshi Kaneko.

**Resources:** Ahmed Hassan Fahal, Emmanuel Edwar Siddig, Takashi Yaguchi.

**Software:** Isato Yoshioka, Yugo Mori.

**Supervision:** Ahmed Hassan Fahal, Satoshi Kaneko, Takashi Yaguchi.

**Validation:** Isato Yoshioka.

**Visualization:** Isato Yoshioka.

**Writing – original draft:** Isato Yoshioka.

**Writing – review & editing:** Ahmed Hassan Fahal, Satoshi Kaneko, Takashi Yaguchi.

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
