## [Decision Letter · Decision Letter 0]

9 Aug 2023

Dear Dr. Yaguchi,

Thank you very much for submitting your manuscript "Specific and sensitive loop-mediated isothermal amplification (LAMP) detection method for Madurella strains, a eumycetoma filamentous fungi causative agent." for consideration at PLOS Neglected Tropical Diseases. As with all papers reviewed by the journal, your manuscript was reviewed by members of the editorial board and by several independent reviewers. The reviewers appreciated the attention to an important topic. Based on the reviews, we are likely to accept this manuscript for publication, providing that you modify the manuscript according to the review recommendations. 

Sincerely,

Joshua Nosanchuk, MD

Section Editor

Reviewer's Responses to Questions

**Key Review Criteria Required for Acceptance?**

**Methods**

-Are the objectives of the study clearly articulated with a clear testable hypothesis stated?

-Is the study design appropriate to address the stated objectives?

-Is the population clearly described and appropriate for the hypothesis being tested?

-Is the sample size sufficient to ensure adequate power to address the hypothesis being tested?

-Were correct statistical analysis used to support conclusions?

-Are there concerns about ethical or regulatory requirements being met?

Reviewer #1: The manuscript by Yoshioka et al., describes the development of a LAMP isothermal amplification system to identify species in the Madurella mycetomatis complex (M. myectomatis, M. pseudomycetomatis, M. tropicana and M. fahalii). Using different primer combinations, they describe LAMP systems that identify (i) M. mycetomatis specifically, (ii) M. falahii specifically, (iii) detect M. mycetomatis, M. tropicana and M. pseudomycetomatis.

The general approach to use of LAMP to identify mycetoma agents is not novel, as a specific LAMP protocol for M. mycetomatis has been published previously. However, the current study adds to the existing knowledge and importantly does permit the detection of the additional species in the M. mycetomatis complex and the specific identification of M. fahalii, which differs from the other organisms in the complex due to its elevated MICs to itraconazole.

On the whole, the data appear robust and support the conclusions made.

My one recommendation would be that it would add to the manuscript to show the electrophoretic analyses of the results of the various LAMP protocols with the different organisms, rather than just the amplification curves. This would be important for centres who might wish to apply the protocol but who only have access to gel electrophoresis as end-point analysis.

Reviewer #2: OK

**Results**

-Does the analysis presented match the analysis plan?

-Are the results clearly and completely presented?

-Are the figures (Tables, Images) of sufficient quality for clarity?

Reviewer #1: The manuscript by Yoshioka et al., describes the development of a LAMP isothermal amplification system to identify species in the Madurella mycetomatis complex (M. myectomatis, M. pseudomycetomatis, M. tropicana and M. fahalii). Using different primer combinations, they describe LAMP systems that identify (i) M. mycetomatis specifically, (ii) M. falahii specifically, (iii) detect M. mycetomatis, M. tropicana and M. pseudomycetomatis.

The general approach to use of LAMP to identify mycetoma agents is not novel, as a specific LAMP protocol for M. mycetomatis has been published previously. However, the current study adds to the existing knowledge and importantly does permit the detection of the additional species in the M. mycetomatis complex and the specific identification of M. fahalii, which differs from the other organisms in the complex due to its elevated MICs to itraconazole.

On the whole, the data appear robust and support the conclusions made.

My one recommendation would be that it would add to the manuscript to show the electrophoretic analyses of the results of the various LAMP protocols with the different organisms, rather than just the amplification curves. This would be important for centres who might wish to apply the protocol but who only have access to gel electrophoresis as end-point analysis.

Reviewer #2: OK

**Conclusions**

-Are the conclusions supported by the data presented?

-Are the limitations of analysis clearly described?

-Do the authors discuss how these data can be helpful to advance our understanding of the topic under study?

-Is public health relevance addressed?

Reviewer #1: (No Response)

Reviewer #2: OK

**Editorial and Data Presentation Modifications?**

Reviewer #1: (No Response)

Reviewer #2: General Comments:

Eumycetoma is a significant public health challenge recognized as a Neglected Tropical Disease. The WHO Fungal Priority Pathogens List (WHO FPPL) released in October 2022 — which should be cited in this paper — has uniquely selected Eumycetoma causative agents under superficial fungal pathogens. Moreover, it ranks them at No.7 within the High Priority Group. The development of molecular biological methods that can simplify and reliably clarify its epidemiology and causative fungal species at the local level is commendable. This paper illustrates the authors' attempt to develop a LAMP identification system for Point of Care Testing using isothermal specific gene amplification for Eumycetoma causative agents. Their efforts to achieve specific DNA amplification for at least the primary causative agents are praiseworthy. However, the value of this test method is limited if it cannot be implemented in Eumycetoma endemic areas; currently, the gene amplification system is only being executed in Japan. Field validation tests are urgently needed. Additionally, the number of positive and negative control fungal species used seems insufficient. Conversely, the descriptions about the amplification system are overly general and verbose, requiring substantial reduction.

Specific Comments:

1 Title: The title reads, "Specific and Sensitive loop-mediated isothermal amplification (LAMP) detection method for Madurella strains, a eumycetoma filamentous fungi causative agent." However, this paper does not detect from clinical or environmental samples but merely identifies from fungal strain DNA using LAMP. The word "detection" may cause confusion and should be removed. A more concise alternative could be: "Specific and Sensitive LAMP method for Madurella strains, a eumycetoma causative agent."

2 Abstract: p.2 L. 28 mentions "Sudan." Eumycetoma is not exclusive to Sudan, and given that the research discussed in this paper isn't region-specific, the wording should be adjusted. Similar changes are needed in the Introduction.

3 Abstract: p.2 L. 32, “(LAMP) detection” should be modified to avoid misunderstanding as the paper validates identification but doesn't discuss detection from various samples. Thus, "detection" should be removed.

4 Abstract: p.3 L. 57, "to detect Madurella strains" should be changed to "to identify Madurella strains."

5 Introduction: p.3 L.68, a reference is required for "(the mycetoma belt)."

6 Fungal Strains: p.5, The number of strains for both Positive and Negative controls is too few. It's acceptable for Madurella mycetomatis with 10 strains and Madurella fahalii with just 4 strains. However, including species like Madurella pseudomycetomatis and Madurella tropicana, which are only studied for one strain, is questionable. The paper should clearly state these limitations. Moreover, many known Eumycetoma causative agents like Falciformispora senegalensis, Curvularia lunata, Scedosporium spp., Zopfia rosatii, Acremonium spp., and Fusarium spp. are listed in the WHO FPPL. The study should consider including them. For differentiation, it should also look into the causative agents of Actinomycetoma.

7 Table 1: p.8, “Chatomium lectangulare” should be corrected to “Chaetomium lectangulare.”

8 Preparation of fungal genomic DNA: p.8, The DNA quantification method is not mentioned, which is crucial for evaluating LAMP detection sensitivity.

9 Table 2 & Fig S4: p. 10 & 35-36, No mention of the PV2 primer set is found in the main text. Based on Fig S4, its specificity is questionable as the Negative Control shows a reaction within 60 minutes. Both should be removed.

10 Design of the primer pairs detecting Madurella spp.: p.12, While Chaetomium spp. is known as a causative agent of Eumycetoma, no evidence of Eumycetoma caused by Chaetomium lectangulare was found. Is it really a causative agent or just used due to sequence similarity? An explanation is needed.

11 Validation of LAMP primers: p.12-13, The discussion here is merely about basic reaction conditions. This section should be simplified, and some of the figures could be removed.

12 LAMP amplification using clinical isolates and other genera: p.13-14, The subtitle "and other genera" seems redundant. Additionally, on p.14 L 221, does "clinical samples" actually mean "clinical isolates"?

13 Table 3: p.14-15, The letter "N" is used in the table, but there's no explanation in the footnotes. If it means "not tested," considering the common fungal species, shouldn't they be tested?

14 Discussion: p15-18, This section is lengthy and should be compressed by at least 66%.

**Summary and General Comments**

Reviewer #1: (No Response)

Reviewer #2: None

PLOS authors have the option to publish the peer review history of their article (what does this mean?). If published, this will include your full peer review and any attached files.

Reviewer #1: No

Reviewer #2: No

Figure Files:

Data Requirements:

Reproducibility:

References

---

## [Editor Report · Decision Letter 1]

7 Sep 2023

Dear Dr. Yaguchi,

Thank you for the rigorous and highly responsive revision of your work. We are pleased to inform you that your manuscript 'Specific and sensitive loop-mediated isothermal amplification (LAMP) method for Madurella strains, eumycetoma filamentous fungi causative agent' has been provisionally accepted for publication in PLOS Neglected Tropical Diseases.

Best regards,

Joshua Nosanchuk, MD

Section Editor

---

## [Editor Report · Acceptance letter]

13 Sep 2023

Dear Dr. Yaguchi,

We are delighted to inform you that your manuscript, "Specific and sensitive loop-mediated isothermal amplification (LAMP) method for *Madurella* strains, eumycetoma filamentous fungi causative agent," has been formally accepted for publication in PLOS Neglected Tropical Diseases.

Best regards,

Shaden Kamhawi

co-Editor-in-Chief

Paul Brindley

co-Editor-in-Chief
